# Corrosion Resistance of Mg/Al Vacuum Diffusion Layers

**Shixue Zhang [1], Yunlong Ding [1,\*], Zhiguo Zhuang [1] and Dongying Ju [2,3,4]**

[1] School of Mechanical Engineering, University of Science and Technology Liaoning, Anshan 114051, China
[2] School of Materials and Metallurgy, University of Science and Technology Liaoning, Anshan 114051, China
[3] Ningbo Haizhi Institute of Materials Industry Innovation, Ningbo 315000, China
[4] Department of Information System, Graduate School of Engineering, Saitama Institute of Technology, Fukaya 369-0293, Japan
\* Correspondence: dylustl@163.com; Tel.: +86-155-4122-8620

**Abstract:** This study used a vacuum diffusion welding process to weld magnesium (Mg1) and aluminum (Al1060). The diffusion layers, with different phase compositions, were separated and extracted by grinding. The diffusion layers' microstructures and phase compositions were analyzed using scanning electron microscopy (SEM) and energy dispersive spectroscopy (EDS). Furthermore, the corrosion resistance of each diffusion layer and the substrates were investigated and compared by performing corrosion immersion tests and linear polarization measurements in a 3.5 wt.% NaCl solution. The results showed that diffusion layers consisting of $Mg_2Al_3$, $Mg_{17}Al_{12}$, and $Mg_{17}Al_{12}$/Mg-based solid solutions were formed at the interface of the Mg1/Al1060 vacuum diffusion joint. Furthermore, each diffusion layer's structure and morphology were of good quality, and the surfaces were free from defects. This result was obtained for a welding temperature of 440 °C and a holding time of 180 min. The corrosion current density of Mg1 was $2.199 \times 10^{-3}$ A/cm$^2$, while that of the Al1060, $Mg_2Al_3$, $Mg_{17}Al_{12}$, and $Mg_{17}Al_{12}$/Mg-based solid solutions increased by order of magnitude, reaching $1.483 \times 10^{-4}$ A/cm$^2$, $1.419 \times 10^{-4}$ A/cm$^2$, $1.346 \times 10^{-4}$ A/cm$^2$, and $3.320 \times 10^{-4}$ A/cm$^2$, respectively. The order of corrosion rate was Mg1 > $Mg_{17}Al_{12}$ and Mg-based solid solution > $Mg_2Al_3$ > $Mg_{17}Al_{12}$ > Al1060. Moreover, all diffusion layers exhibited an improved corrosion resistance compared to Mg1. This was especially the situation for the $Mg_2Al_3$ layer and $Mg_{17}Al_{12}$ layer, whose corrosion resistances were comparable to that of Al1060.

**Keywords:** Mg1; Al1060; diffusion welding; intermetallic compounds; diffusion layers; corrosion resistance

## 1. Introduction

With the continuous and rapid development of modern industrial technology, the sustainable development of lightweight materials for environmental protection and energy savings has attracted more and more attention in today's society [1]. Magnesium and aluminum are two lightweight nonferrous metals of low density and high specific strength compared to common structural materials such as steel. Their alloys have been widely used in global transportation, especially in the automotive and aerospace industries [2–4]. At present, rolling, bonding, and welding are commonly used processes by which it is possible to achieve an effective combination of Mg and Al heterogeneous metals. These processes not only result in an optimization of the structural qualities but also take full advantage of the respective metals' properties [5,6]. The welding methods that are used at present in realizing the combination of Mg/Al heterogeneous metals mainly include laser welding [7], TIG welding [8], stir friction welding [9,10], diffusion welding [11,12], and ultrasonic welding [13]. The chemical reactivities of magnesium and aluminum are relatively high, and defects such as oxidation, cracks, and pores can easily appear in the process of traditional fusion welding. Vacuum diffusion welding is a solid-state welding method that uses a low heat input during the welding process. As a result, the metal base material does not melt; it only undergoes a microscopic plastic deformation at the surface.

The quality of the welded joint is, therefore, relatively stable. This method is suitable for welding metal materials with different physical properties, such as the coefficient of thermal expansion. It is, thus, applicable to magnesium and aluminum.

Corrosion is the physical and chemical reaction between materials and their surroundings, which changes the properties of a material. Corrosion is harmful to the production and development of today's industry. Metal corrosion significantly reduces its service life and easily causes potential safety hazards using precision fields [14,15]. It is well known that magnesium and magnesium alloys experience poor corrosion resistance and are susceptible to corrosion in neutral solutions containing chloride ions. Poor corrosion resistance is a crucial bottleneck that limits their wide application in areas with high safety requirements [16]. However, aluminum and aluminum alloys exhibit an improved corrosion resistance relative to magnesium and magnesium alloys in chlorine-containing environments [17]. Intermetallic compounds ($Mg_2Al_3$, $Mg_{17}Al_{12}$) are formed in the joint during vacuum diffusion welding of the Mg/Al pair, thereby forming a continuous diffusion layer. Numerous studies have shown that Mg/Al intermetallic compounds are hard and brittle, and their wide distribution at the common interface leads to further deterioration of the mechanical properties of the joint [18–20]. Therefore, in the vacuum diffusion welding of dissimilar metals (e.g., the Mg/Al pair), the primary research has focused on effectively suppressing the formation of brittle intermetallic compounds. Many researchers have extensively investigated the optimization of welding process parameters and the possibility of an implantation of a suitable interlayer to suppress the formation of intermetallic compounds [21–23]. No matter what method is used, the generation of an intermetallic compound cannot be avoided entirely. It is only possible to control its formation, or change its distribution, to a certain extent in the strives to improve the performance of the joint. The generation of intermetallic compounds in joints is inevitable [24–26]. Therefore, under the conditions of the overall controlled corrosion resistance of Mg and Al, it is essential to study the corrosion behavior of the Mg/Al vacuum diffusion composite plates and the intermetallic compounds at the joint. Related workers had made some reports on the corrosion resistance of magnesium-aluminum intermetallic compounds. The research of Zhang et al. [27] showed that the magnesium aluminum intermetallic compound coating could significantly improve the wear resistance and corrosion resistance of magnesium substrate and protect magnesium substrate from wear and corrosion. Bu et al. [28] successfully deposited intermetallic compound $Mg_{17}Al_{12}$ particle reinforced pure Al coatings onto AZ91D magnesium substrate, their measured potentiodynamic polarization curves in 3.5 wt.% NaCl solution showed that the corrosion current density of the magnesium substrate decreased by more than one order of magnitude after the deposition of the coating.

However, to our knowledge, no reports deal with the corrosion resistance evaluation of the Mg/Al vacuum diffusion composite plates, nor with diffusion layers, when immersed in an aggressive NaCl environment. In the present study, the Mg/Al pair welding has been the first one using a vacuum diffusion welding technique. The microstructure and phase composition of the diffusion layers were observed and analyzed by Scanning Electron Microscopy (SEM) and energy dispersive spectroscopy (EDS). Each diffusion layer was separated and extracted. Furthermore, the corrosion mechanism was analyzed using the optical microscope (OM), SEM, and EDS. This was also the situation for the Mg/Al vacuum diffusion composite plates and the different diffusion layers. This paper separated diffusion layers of Mg/Al and their corrosion behavior in a 3.5 wt.% NaCl solution was studied, which was nearly the first try. The purpose of the present study was not only to explore the corrosion resistances of the diffusion layers and the Mg/Al matrix, but also to lay the theoretical foundation for an improvement of the corrosion resistance of magnesium by using alloying aluminum material.

## 2. Experimental Materials and Methods

### 2.1. Materials

The base metals selected for the experiments were aluminum (Al1060) and magnesium (Mg1), and the chemical composition of these metals is listed in Tables 1 and 2, respectively. The specimens used for the vacuum diffusion welding test were of size 80 mm × 20 mm × 3 mm.

**Table 1.** Chemical composition of Mg1 (mass fraction, %).

| Mg | Al | Mn | Cu | Si | Fe | Ca | Ni |
|----|----|----|----|----|----|----|----|
| Bal. | 0.2 | 0.22 | 0.0008 | 0.012 | 0.0021 | 0.0015 | 0.0009 |

**Table 2.** Chemical composition of Al1060 (mass fraction, %).

| Al | Mg | Mn | Cu | Si | Fe | Zn | Ni |
|----|----|----|----|----|----|----|----|
| Bal. | 0.05 | 0.10 | 0.007 | 0.012 | 0.20 | 0.25 | 0.0016 |

### 2.2. Preparation of Mg1/Al1060 Vacuum Diffusion Layers

Before welding, the metal surfaces were ground with 400#, 800#, 1200#, 1500#, and 2000# abrasive papers to remove oxide films from these surfaces. After that, it was polished using a polishing cloth and a diamond abrasive paste. Ultrasonic cleaning was performed after the polishing, and absolute ethanol was used to remove any impurities or oil stains from the surface. As the next step, the treated base material was put into a unique mold with a "magnesium on top of aluminum" stacking and placed in a vacuum heating furnace for welding. Based on the phase diagram of the Mg/Al binary alloy, the welding temperature was chosen to be 440 °C. Considering that the Mg/Al vacuum diffusion layers will gradually grow with an extension of the welding time, the holding time during the welding process was selected as 180 min. This time ensures that the diffusion reactions were entirely carried out and that a diffusion layer with sufficient thickness was obtained. This thickness should facilitate the subsequent extraction of diffusion layers with different phase compositions and be sufficiently large for the following corrosion experiments. In order to avoid a thermal shock that would affect the quality of the welded joint, the sample was slowly cooled to room temperature within the furnace after completion of the welding. The heating rate during the welding process was 10 °C/min, and the vacuum pressure was less than $1 \times 10^{-2}$ Pa. The welding process flow is demonstrated in Figure 1.

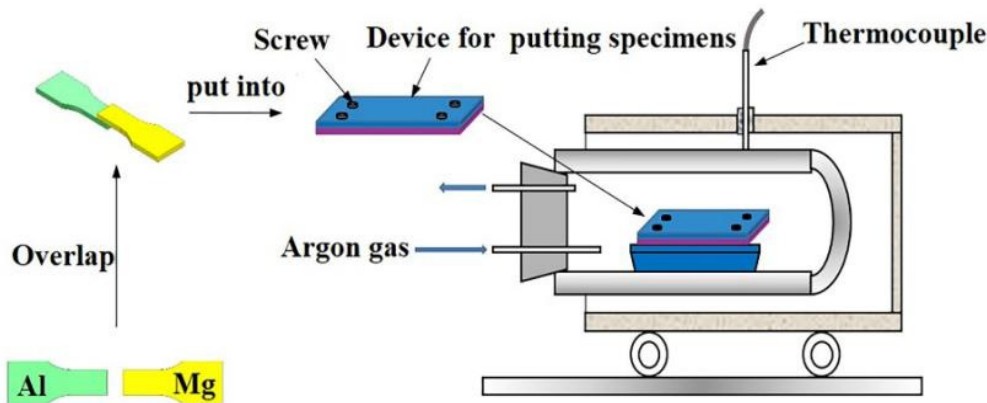

**Figure 1.** Flow chart of the vacuum diffusion welding process.

### 2.3. Assessment of Corrosion Resistance

The interface of a Mg1/Al1060 welded joint was sampled by using a wire-cutting technique. The corrosion resistance of these samples (of size 20 mm in length and 8 mm in width) was characterized by corrosion immersion and linear polarization techniques.

The tests were repeated three times under each condition to ensure accuracy and avoid accidental errors. Before these analyses, all samples were polished to 2000 grit and cleaned with anhydrous ethanol. All corrosion resistance tests were carried out in 3.5 wt.% NaCl solution under atmospheric conditions.

### 2.3.1. Corrosion Immersion Tests

The polished Mg1/Al1060 composite plates were immersed in a 3.5 wt.% NaCl solution and removed every 90 min. During these interruptions, the corrosion products on the surface were removed using ultrasonic cleaning with absolute ethanol, and the plates were then dried with a hair dryer. The evolution of the surface morphologies could, in this way, be followed by using an optical microscope (Thermo Fisher Scientific, Waltham, MA, America).

### 2.3.2. Linear Polarization

The wire-cut specimens used for linear polarization tests were encapsulated with tooth-powder. The purpose was to expose the diffusion layers both uniformly and continuously. They were, after that, ground from the magnesium substrate side with a metallographic polishing machine. When the grinding was close to the exposure of diffusion layers, they were carefully polished with the polishing cloth and abrasive paste instead of abrasive papers to prevent the removal of abrasive papers being too large to get a single-component diffusion reaction layer; the EDS (Thermo Fisher Scientific, Waltham, MA, America) was used to verify this.

Linear polarization measurements were used for a more quantitative assessment of the corrosion resistance of the diffusion layers. These measurements were carried out for both Mg1 and Al1060 using a CS Instruments Electrochemical Work Station (CS350, Wuhan, China). A comparison between Mg1 and Al1060 was then easily achieved. A conventional three-electrode electrochemical cell setup was employed, which consisted of the test sample as the working electrode (with an exposure area of 1.6 cm$^2$), a silver chloride electrode as the reference electrode, and a platinum electrode as the counter electrode. The linear polarization measurements were then made by applying a potential in the range of $-300$ mV to $+300$ mV, with a scan rate of 1 mV/s. Linear polarization curves were obtained, and the electrochemical measurements were completed. Polarization data, including the corrosion potential ($E_{corr}$), corrosion current density ($I_{corr}$), polarization resistance ($R_p$), and corrosion rate ($V_{cor}$), could be deduced from the linear polarization curves (i.e., log I vs. E plot). Furthermore, the $I_{corr}$ values were obtained from the intersection of the Tafel slope. Moreover, the $R_p$ and Vcor values were calculated using the Corrview 3.10 software, which was provided by the electrochemical workstation (CS350). Furthermore, the samples' surface morphology was studied using SEM and EDS.

## 3. Results

### 3.1. Microstructure and Phase Composition of the Mg1/Al1060 Layers

Figure 2 shows the microstructure of the Mg1/Al1060 joint, which was formed at a welding temperature of 440 °C and for a holding time of 180 min. It is clear that the joint is well combined, and there are no defects such as holes, burning, or incomplete fusion. Furthermore, the initial interface between the Mg1 and Al1060 substrates disappeared, and a diffusion layer was formed at the joint position. An enlarged view of the diffusion layer morphology can be seen in Figure 2b, which clearly shows that the diffusion layer is composed of three layers. The organizational structure of layer 1 is relatively uniform, while layer 2 has an irregular columnar structure towards the Mg1 substrate. Moreover, layer 3 consists of a relatively homogeneous eutectic structure and is much thicker than layers 1 and 2. The elemental compositions of these diffusion layers were analyzed using an EDS point scan. The results are shown in Figure 3. The elemental composition of point A in layer 1 was composed of approximately 40% Mg and 60% Al. According to the analysis in Ref. [1] and the Mg/Al alloy phase diagram, the composition of point A was the $Mg_2Al_3$ phase. Furthermore, point B of layer 2 consisted of nearly 40% Al and 60% Mg, which

suggests that layer 2 was composed of the $Mg_{17}Al_{12}$ phase. Figure 3c shows the EDS result from the light point C of layer 3. This point consists of nearly 40% Al and 60% Mg, which was determined to be the $Mg_{17}Al_{12}$ phase. Additionally, Figure 3d shows the EDS result from the dark point D of layer 3, where Mg has increased to nearly 90%, while Al has decreased to nearly 10%. Thus, this point did mainly consist of a Mg-based solid solution. Hence, it was demonstrated that the eutectic structure in layer 3 consisted of both $Mg_{17}Al_{12}$ and a Mg-based solid solution. Based on these results, it can be understood that the order of diffusion layers from the Al side to the Mg side was: $Mg_2Al_3$ layer, $Mg_{17}Al_{12}$ layer, $Mg_{17}Al_{12}$/Mg-based solid solution layer.

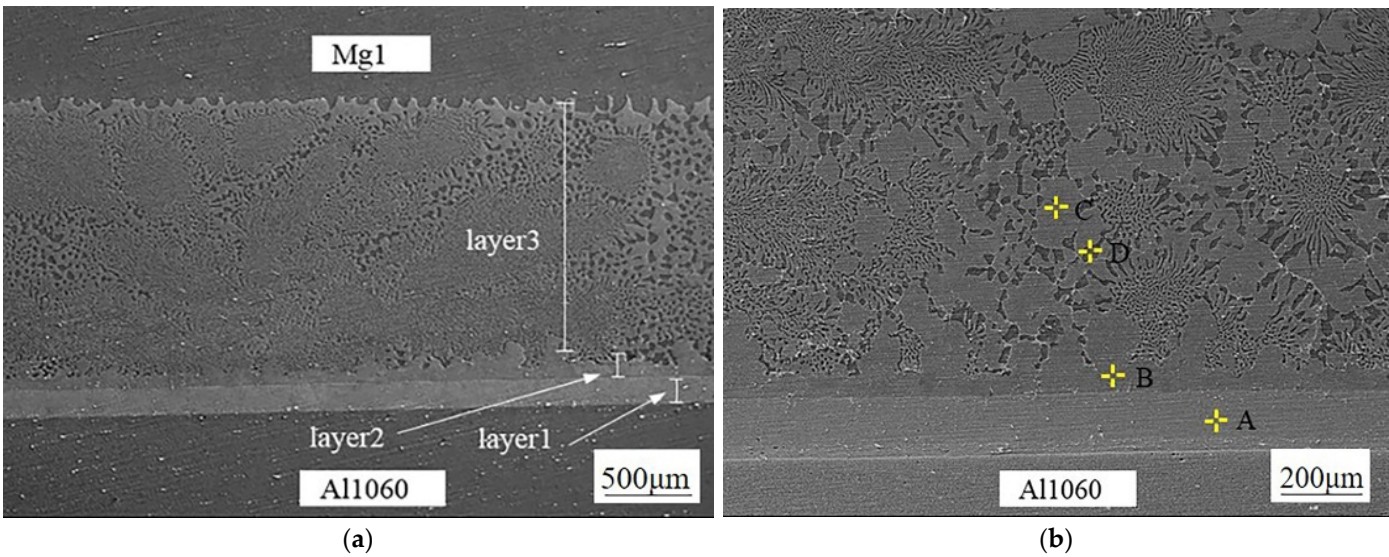

**Figure 2.** SEM images of the Mg1/Al1060 diffusion layers: (**a**) SEM image at low magnification, (**b**) SEM image at high magnification.

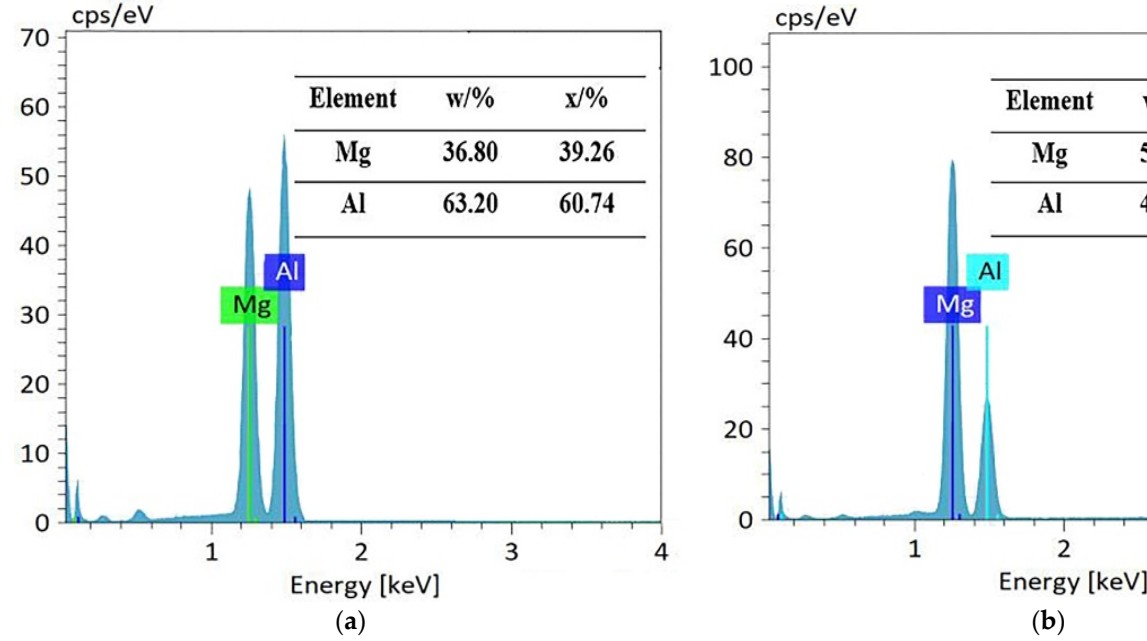

**Figure 3.** *Cont.*

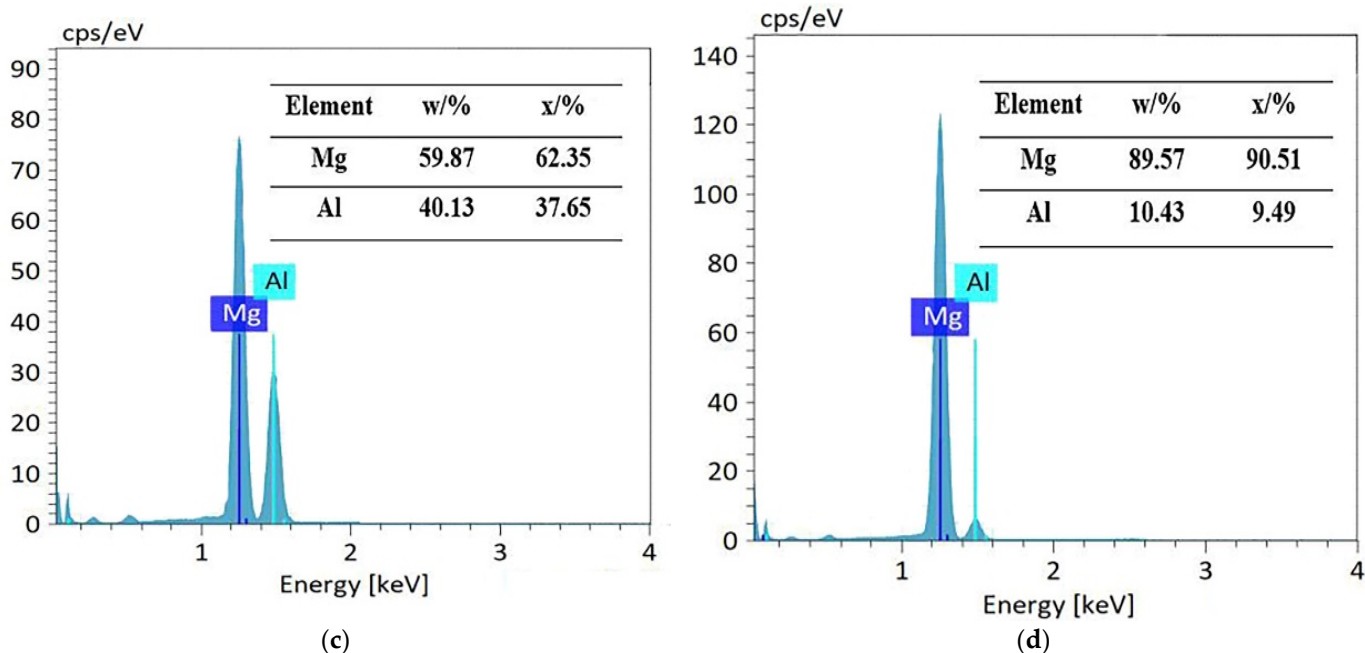

**Figure 3.** Elemental composition in different regions of the diffusion layer: (**a**) point A, (**b**) point B, (**c**) point C, and (**d**) point D.

*3.2. Corrosion Immersion Test Results*

Figure 4 shows a significant surface morphology evolution with time for a Mg1/Al1060 composite plate immersed in a 3.5 wt.% NaCl solution. It can be seen that the surface morphology of the diffusion layers and the two substrates was of high quality before any soaking had taken place. Thus, there were no apparent defects. On the contrary, the Mg1 substrate became severely degraded after 90 min of immersion in a 3.5 wt.% NaCl solution. A large piece of the Mg1 metal was corroded away from the composite plate, and the surface of the remaining part turned black (from a metallic luster). However, the surfaces of the diffusion layers and the Al1060 substrate showed no obvious corrosion defects. They had, thus, stronger corrosion resistance than Mg1. Furthermore, the surface of the Mg1 substrate became coarser and darker upon further immersion in the NaCl solution. In addition, typical pitting corrosion occurred on the surface of the Al1060 substrate, and these pits were evenly distributed on the surface. However, there were no noticeable changes on the surfaces of the diffusion layers (see Figure 4c). For even longer immersion times, Mg1 became entirely corroded, and the corrosion of Al1060 worsened. In addition, the pits on the Al1060 surface were gradually enlarged. Interestingly, there were still no evident corrosion-related defects on the surfaces of the diffusion layers (see Figure 4d). When the immersion time reached 360 min, the pitting corrosion on the Al1060 surface became more significant. The pits increased in quantity and gradually appeared as a honeycomb distribution. Some parts of the diffusion layers were also corroded off; traces of pits were left on its surface, with a more pronounced concentration on the $Mg_{17}Al_{12}$ and Mg-based solid solution layer. When the immersion time reached 450 min, the original metal surface of Al1060 was almost completely destroyed. One can also find that the corrosion pits had grown even further and connected in a continuous wave-like pattern. At the same time, the corrosion pits on the diffusion layers had increased in number and significantly enlarged. This result proves that the degree of corrosion became severe with an increase in immersion time.

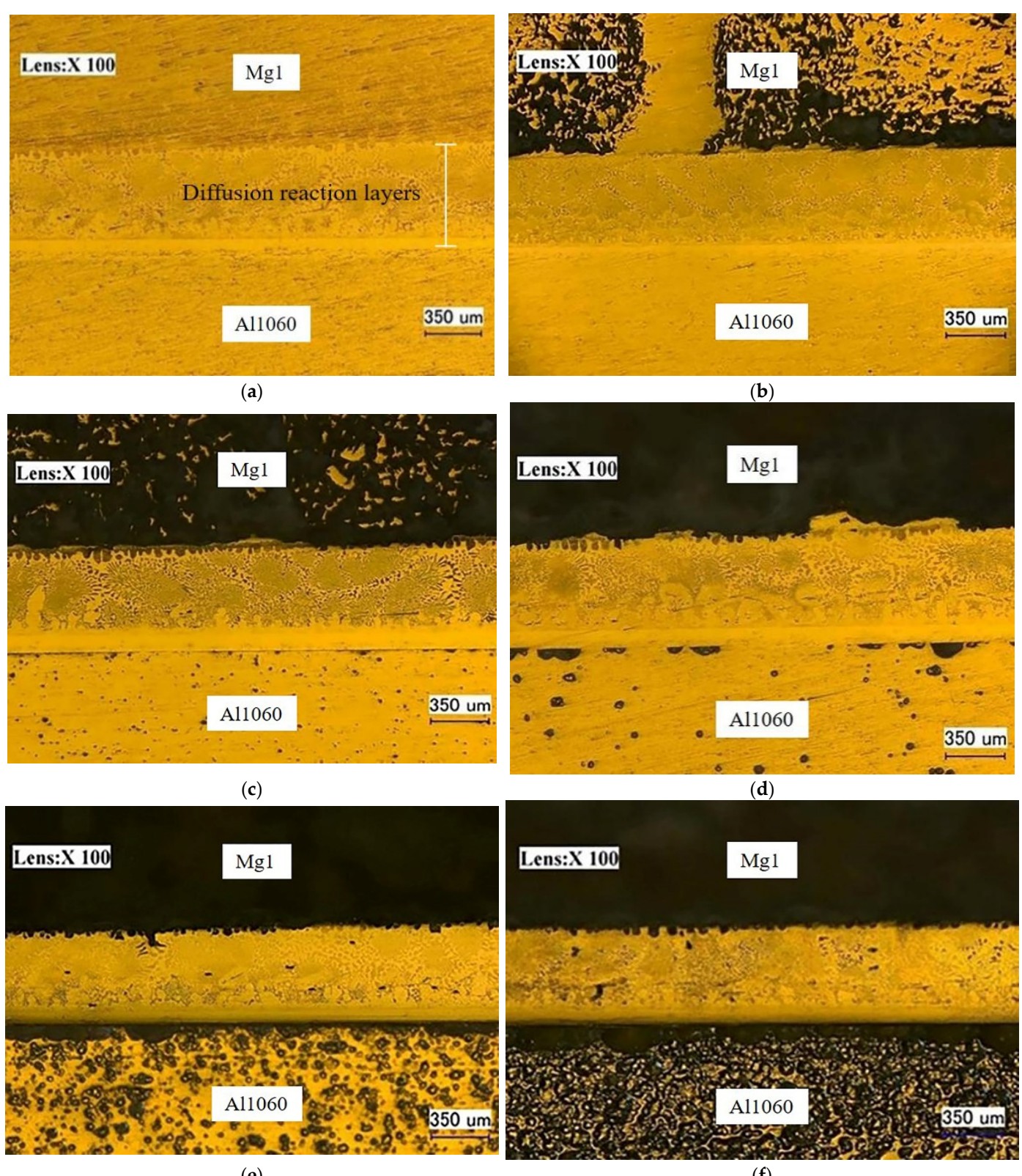

**Figure 4.** Surface morphology of the substrates and diffusion layers for different immersion times: (**a**) 0 min, (**b**) 90 min, (**c**) 180 min, (**d**) 270 min, (**e**) 360 min, and (**f**) 450 min.

By soaking in a 3.5 wt.% NaCl solution, it could initially be seen that the Mg1 substrate was most easily corroded in the Mg1/Al1060 composite plates. Many studies have confirmed that the oxide films formed on the surfaces of magnesium and magnesium alloys in the air cannot effectively protect the surfaces in a solution containing Cl ions. Rapid corrosion occurs, which is due to the low electrode potential of magnesium. Thus, magnesium will act as an anode when it is in contact with other alloys, causing electrochemical corrosion. This explains the present study's rapid corrosion and degradation of the Mg1 substrate. On the other hand, Al1060 showed a stronger corrosion resistance than Mg1, with typical pitting corrosion occurring after immersion. The continuous pitting corrosion was found to penetrate deep into the metal, causing severe damage to the Al1060 substrate. However, the diffusion layers showed the most negligible corrosion. The explanation is their direct contact with the Mg1 substrate, where Mg1 acts as an anode. Thus, the Mg1 substrate underwent rapid galvanic corrosion, which indirectly protected the diffusion layer. As more and more magnesium substrate was corroded, the protective effect weakened, and the reaction layers gradually corroded. Therefore, the diffusion layers were the least corroded, with corrosion defects gradually appearing on the surface after an immersion time of 360. It was observed that the corrosion defects were mainly concentrated on the $Mg_{17}Al_{12}$ and Mg-based solid solution diffusion layer.

### 3.3. Cross-Sectional Structure and Energy Spectrum Analysis

Figure 5 shows the cross-sectional structure of each diffusion layer ($Mg_2Al_3$ layer, $Mg_{17}Al_{12}$ layer, and $Mg_{17}Al_{12}$ and Mg-based solid solution layer) after grinding. It can be seen that the structure and morphology of each diffusion layer are intact, and the surface is free from defects. In order to verify the composition of each diffusion layer, its composition was examined by using EDS. Figure 6 shows the elemental detection results of the EDS surface scan, where red represents the Mg element and green represents the Al element. It can be seen that the Mg and Al elements were uniformly distributed in these samples, which reflects the presence of only one single and homogeneous phase in each of these diffusion layers. The Al element, as represented by green in the $Mg_2Al_3$ diffusion layer, is dominating in the samples. In comparison, the presence of the Mg element, as represented by red, has significantly increased in the $Mg_{17}Al_{12}$ diffusion layer. While the most apparent Mg element can be observed in $Mg_{17}Al_{12}$ and Mg-based solid solution diffusion reaction layer. These experimental findings are consistent with the theoretical composition of phases in each diffusion layer. The chemical composition of position A in the $Mg_2Al_3$ layer, position B in the $Mg_{17}Al_{12}$ layer, and position C in the $Mg_{17}Al_{12}$ and Mg-based solid solution layer has been analyzed by performing an EDS area scan. The results are presented in Table 3, where it can be seen that the content of aluminum has gradually decreased, while the content of magnesium has gradually increased when going from box A to box C. This result further verifies the phase composition of the extracted diffusion layers in combination with the Mg/Al alloy phase diagram.

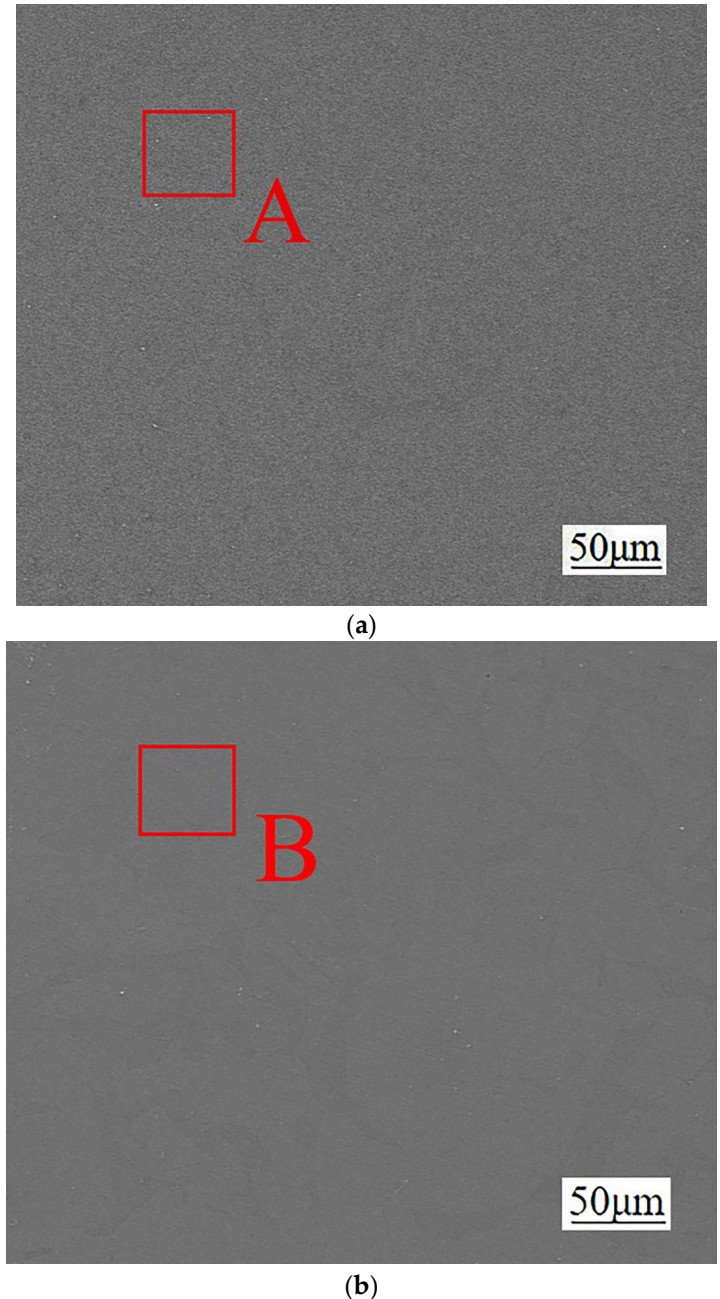

(**a**)

(**b**)

**Figure 5.** *Cont.*

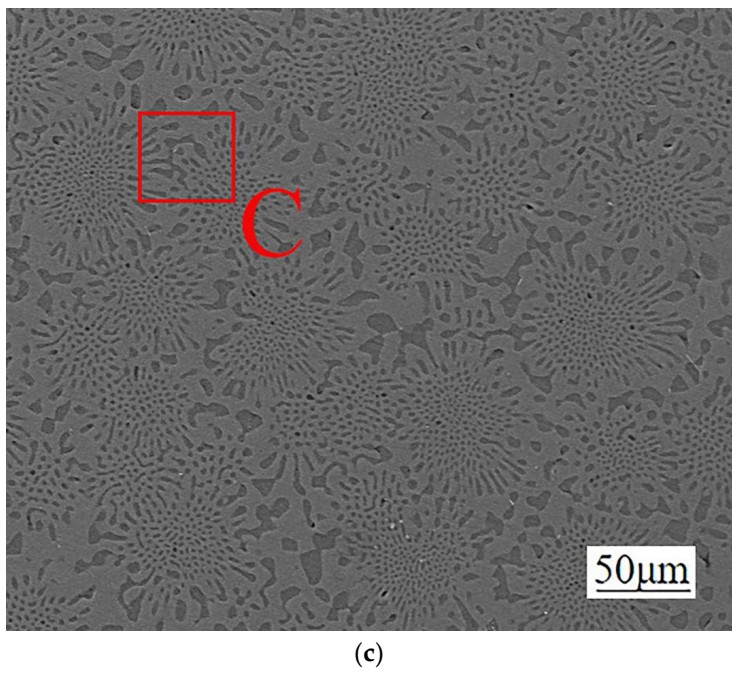

(**c**)

**Figure 5.** Cross-sectional structure of different diffusion layers: (**a**) Mg$_2$Al$_3$ layer, (**b**) Mg$_{17}$Al$_{12}$ layer, and (**c**) Mg$_{17}$Al$_{12}$ and Mg-based solid solution layer.

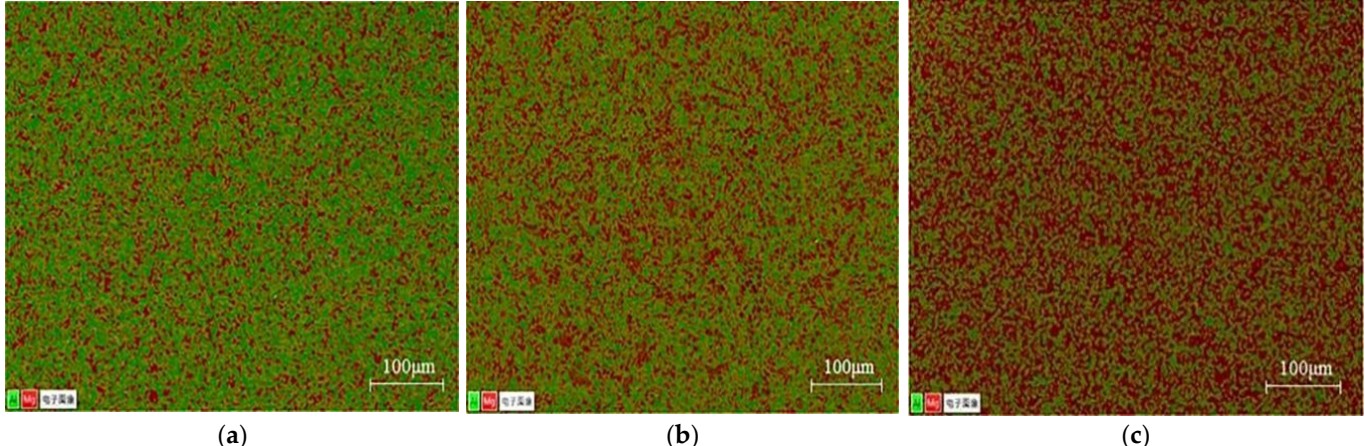

(**a**)            (**b**)            (**c**)

**Figure 6.** EDS surface scanning analysis of different diffusion layers: (**a**) Mg$_2$Al$_3$ layer, (**b**) Mg$_{17}$Al$_{12}$ layer, and (**c**) Mg$_{17}$Al$_{12}$ and Mg-based solid solution layer.

**Table 3.** EDS results of the distinct regions presented in Figure 5.

| Position | Mole Fraction/% | |
|:--------:|:---------------:|:----:|
| | **Al** | **Mg** |
| A | 61.60 | 38.40 |
| B | 39.72 | 60.28 |
| C | 26.14 | 73.86 |

*3.4. Linear Polarization*

3.4.1. Analyses of Polarization Curves

After the immersion in a 3.5 wt.% NaCl solution, Figure 7 shows the polarization curves for the Mg1 and Al1060 substrates. It also shows the polarization curves for the Mg$_2$Al$_3$, Mg$_{17}$Al$_{12}$, Mg$_{17}$Al$_{12}$, and Mg-based solid solution layers. It is clear that the cathodic reaction is most prominent for the Mg1 substrate, while Al1060, Mg$_2$Al$_3$, and

$Mg_{17}Al_{12}$ show a passive and similar electrochemical behavior over a wide potential range. This result suggests that the corrosion resistance of Al1060 and each of the diffusion layers are better than the corrosion resistance of Mg1. Based on the polarization curves, polarization data (including corrosion potential ($E_{corr}$), corrosion current density ($I_{corr}$), polarization resistance ($R_p$), and corrosion rate ($V_{cor}$)) have been calculated and are listed in Table 4.

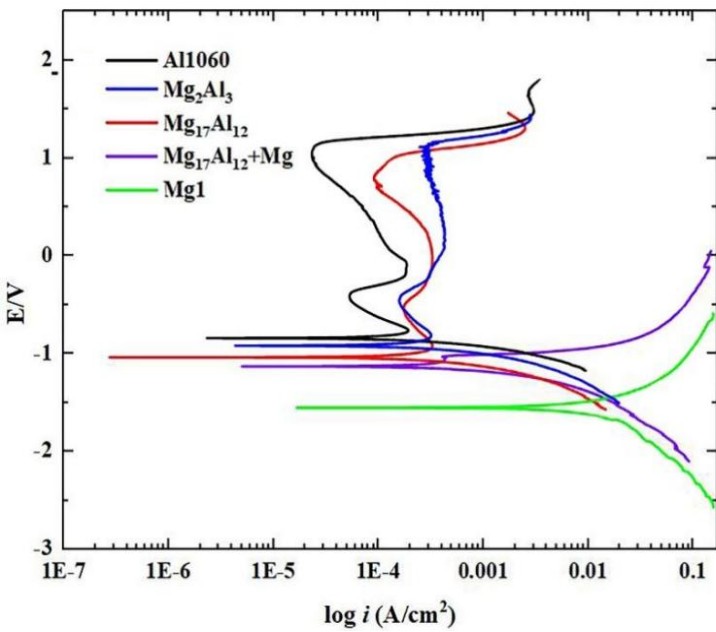

**Figure 7.** Electrochemical polarization curves of the substrates and different diffusion layers in a 3.5 wt.% NaCl solution.

**Table 4.** Polarization data of Al1060, Mg1, and diffusion layers in 3.5 wt.% NaCl solution.

| Samples | $E_{corr}$ (V) | $I_{corr}$ (A/cm$^2$) | $R_p$ ($\Omega$ cm$^2$) | $V_{cor}$ (mm/year) |
|---|---|---|---|---|
| Al1060 | −0.84 | $1.483 \times 10^{-4}$ | 175.96 | 5.0377 |
| $Mg_2Al_3$ | −0.98 | $1.419 \times 10^{-4}$ | 183.9 | 7.2302 |
| $Mg_{17}Al_{12}$ | −1.03 | $1.346 \times 10^{-4}$ | 209.3 | 6.3529 |
| $Mg_{17}Al_{12}$ + Mg | −1.14 | $3.320 \times 10^{-4}$ | 81.007 | 16.414 |
| Mg1 | −1.55 | $2.199 \times 10^{-3}$ | 11.861 | 115.97 |

It can be seen that the corrosion potentials of the Al1060 substrate and the $Mg_2Al_3$ and $Mg_{17}Al_{12}$ layers are very similar, and their absolute values are relatively small. On the contrary, the corrosion potential of the Mg1 substrate is much smaller than those of Al1060 and each diffusion layer. Its absolute value is also the largest, which implies a poor corrosion resistance of Mg1. The order of corrosion current densities for the five tested samples were: Mg1 > $Mg_{17}Al_{12}$ and Mg-based solid solution > Al1060 > $Mg_2Al_3$ > $Mg_{17}Al_{12}$. The corrosion current density was determined by the dissolution degree of the material. The higher the corrosion current density, the smaller the charge transfer resistance, which means that the material's corrosion resistance is weaker [29]. Furthermore, the Al1060 substrate, and each diffusion layer, showed a reduction of one order for the Icorr value as compared with Mg1 ($2.199 \times 10^{-3}$ A/cm$^2$). This is a reflection of the much slower degradation of the Al1060 substrate and each diffusion layer as compared with Mg1. Additionally, the polarization resistance ($R_p$) of the Mg1 substrate was 11.861 $\Omega$ cm$^2$, while $Mg_{17}Al_{12}$ and the Mg-based solid solution layer showed a significant increase (81.007 $\Omega$ cm$^2$). This indicates a marginal improvement in corrosion resistance for $Mg_{17}Al_{12}$ and the Mg-based solid solution layer, while the improvements for the Al1060 substrate and the $Mg_2Al_3$ and $Mg_{17}Al_{12}$ layers were more obvious. The $R_p$ values increased to 175.96 $\Omega$ cm$^2$, 183.9 $\Omega$ cm$^2$, and 209.3 $\Omega$ cm$^2$, respectively, demonstrating that the corrosion resistance of these three

samples increased significantly compared to Mg1.It could also be seen that the corrosion rate of Mg1 had the highest value of 115.97 mm/year. Thus, the quality of the sample would be greatly affected if exposed to a corrosive environment for too long.

The above results show that the Mg1 substrate had the worst corrosion resistance in an aggressive NaCl environment, followed by the $Mg_{17}Al_{12}$ and Mg-based solid solution layer. Furthermore, the $Mg_2Al_3$ and $Mg_{17}Al_{12}$ layers showed a similar corrosion resistance as the Al1060 substrate, which was significantly better than the corrosion resistance of the Mg1 substrate. In the study of Song et al. [30,31], it was found that the $Mg_{17}Al_{12}$ phase can act as an anode barrier to restrain the overall corrosion of magnesium alloy; the $Mg_{17}Al_{12}$ phase was inert to corrosion in the solution containing $cl^-$, and a passive regions table over a similar potential range was observed in the polarization curve of $Mg_{17}Al_{12}$ measured by them. Many researchers have used the idea of alloying in the preparation of Mg/Al intermetallic compound coatings on the surface of magnesium alloys. The purpose is to improve the corrosion resistance of magnesium alloys [32–34]. Ji et al. [32] modified the pure aluminum coating deposited on the AZ91D substrate by friction stir-spot-processing. They observed that the $Mg_2Al_3$ and $Mg_{17}Al_{12}$ phases were irregularly distributed in the coating after modification, and the corrosion current density of the cold-sprayed Al coating was remarkably reduced. Irregularly distributed intermetallic compounds in the coating lead to the significantly enhanced corrosion resistance of the AZ91D substrate. Moreover, the results in the present study strongly agree with the results from the potentiodynamic polarization experiment proposed by Spencer K et al. [34]. This study aimed at Mg/Al vacuum diffusion layers. Each diffusion layer was extracted separately, and the difference in corrosion resistance between different types of Mg/Al intermetallic compounds and the base was investigated in more detail. The results of the corrosion resistance polarization curves also show consistency with the above corrosion immersion tests.

### 3.4.2. Analysis of Corrosion Morphology

Figure 8 shows the surface morphologies of the Al1060 and Mg1 substrates and of the $Mg_2Al_3$, $Mg_{17}Al_{12}$, and $Mg_{17}Al_{12}$ and Mg-based solid solution layers after potentiostatic electrochemical measurements. The overall surface compositions were measured by using EDS, and the results are shown in Figure 9. It can be seen that noticeable corrosion pits were formed on the surface of the Al1060 substrate. Some of these corrosion pits appeared to sparkle, indicating that there were corrosion products in the vicinity of the pits (see Figure 8a). However, the original surface of the Mg1 substrate has been severely destroyed (see Figure 8b). The corrosion on this surface was much more severe, with corrosion cracks and pits continuously distributed all over the surface. Moreover, the corrosion products formed on the surface were more compact than those formed on the Al1060 substrate and on each of the diffusion layers. Pitting corrosion occurs in the $Mg_2Al_3$ layer and $Mg_{17}Al_{12}$ layer likewise (Figure 8c,d). A few pits and cracks could be observed on the surfaces of the $Mg_2Al_3$ and $Mg_{17}Al_{12}$ layers, even though the whole surfaces were relatively smooth. Corrosion products were distributed over the surface in the form of needles and blocks. While the corrosion was more severe for the $Mg_{17}Al_{12}$ and Mg-based solid solution layer, as compared with $Mg_2Al_3$ and $Mg_{17}Al_{12}$, the surfaces suffered from corrosion damage with typical pitting and localized corrosion characteristics. Obvious cracks and giant corrosion pits were thus formed on its surfaces (see Figure 8f).



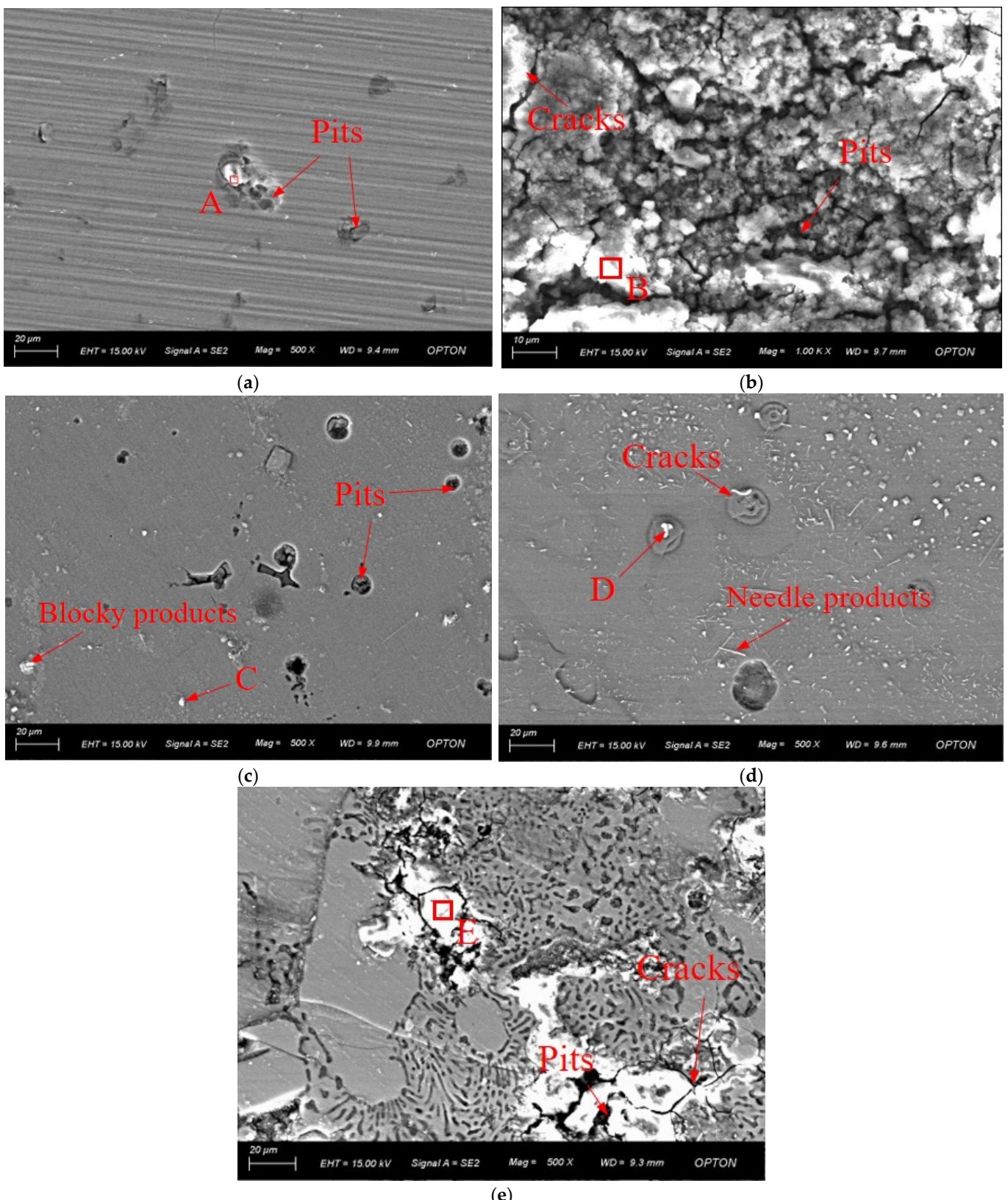

**Figure 8.** Surface morphologies of the substrates and different diffusion layers after potentiodynamic electrochemical measurements: (**a**) Al1060, (**b**) Mg1, (**c**) $Mg_2Al_3$ layer, (**d**) $Mg_{17}Al_{12}$ layer, and (**e**) $Mg_{17}Al_{12}$ and Mg-based solid solution layer.

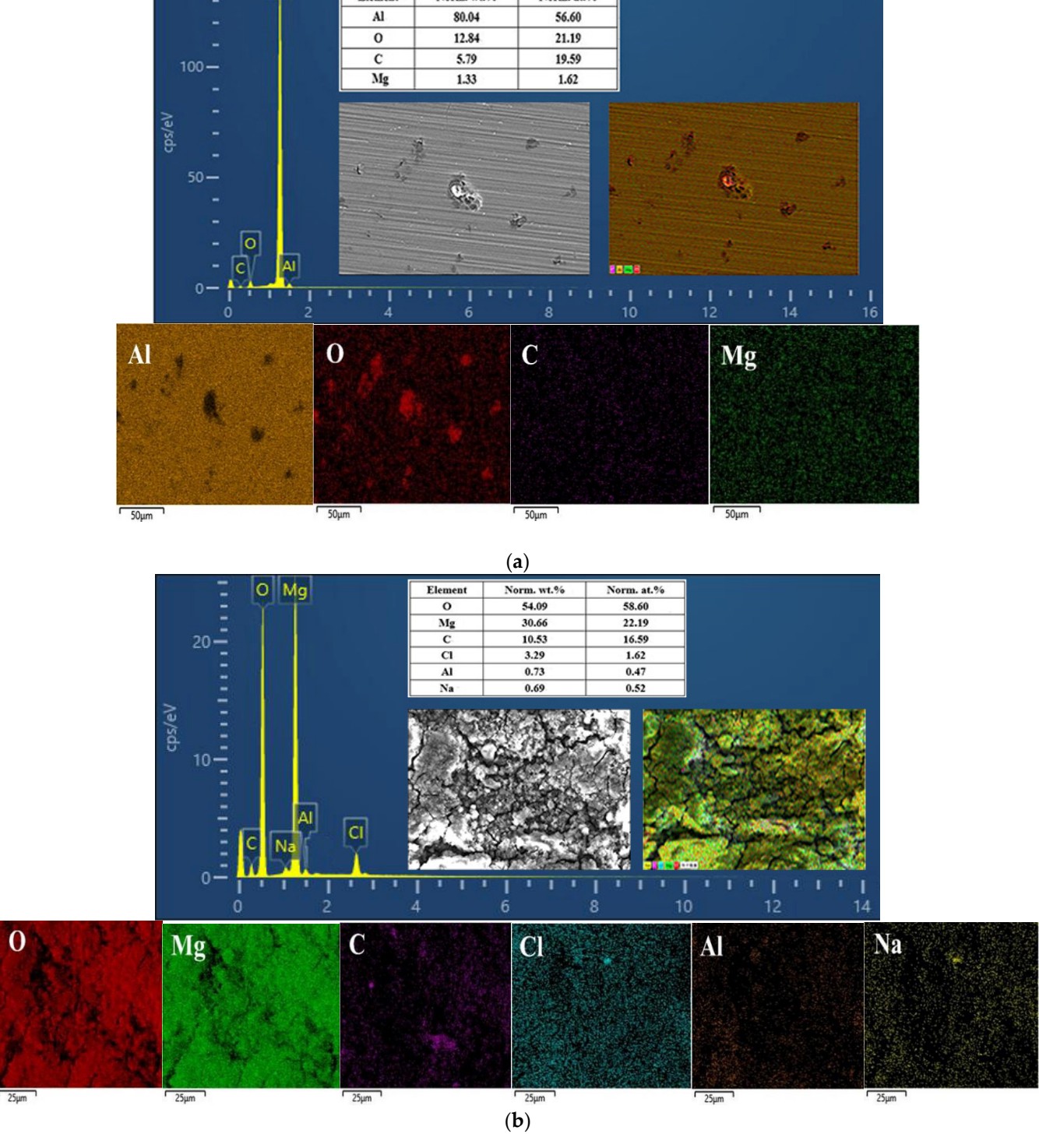

(**a**)

(**b**)

**Figure 9.** *Cont.*

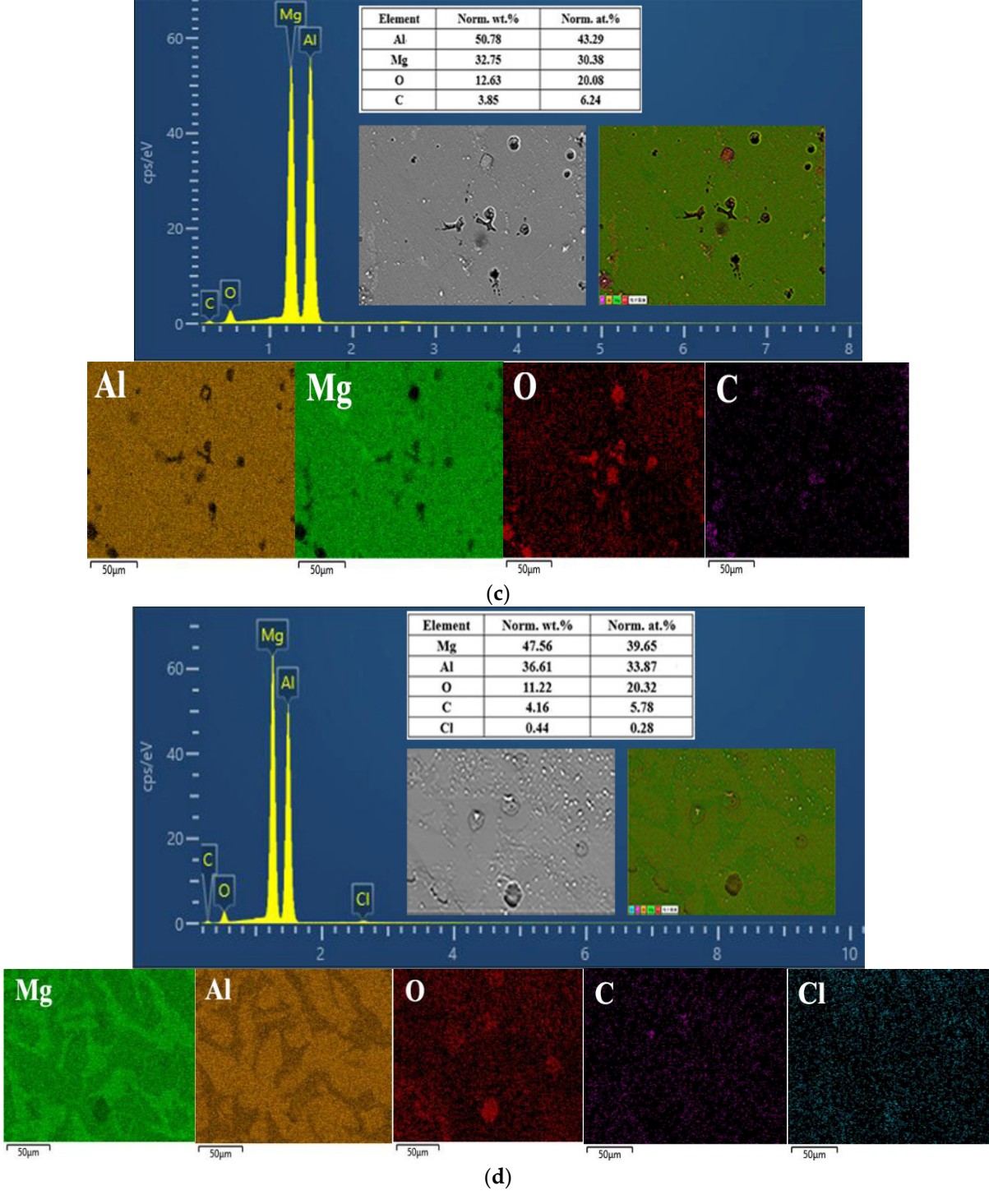

**Figure 9.** *Cont.*

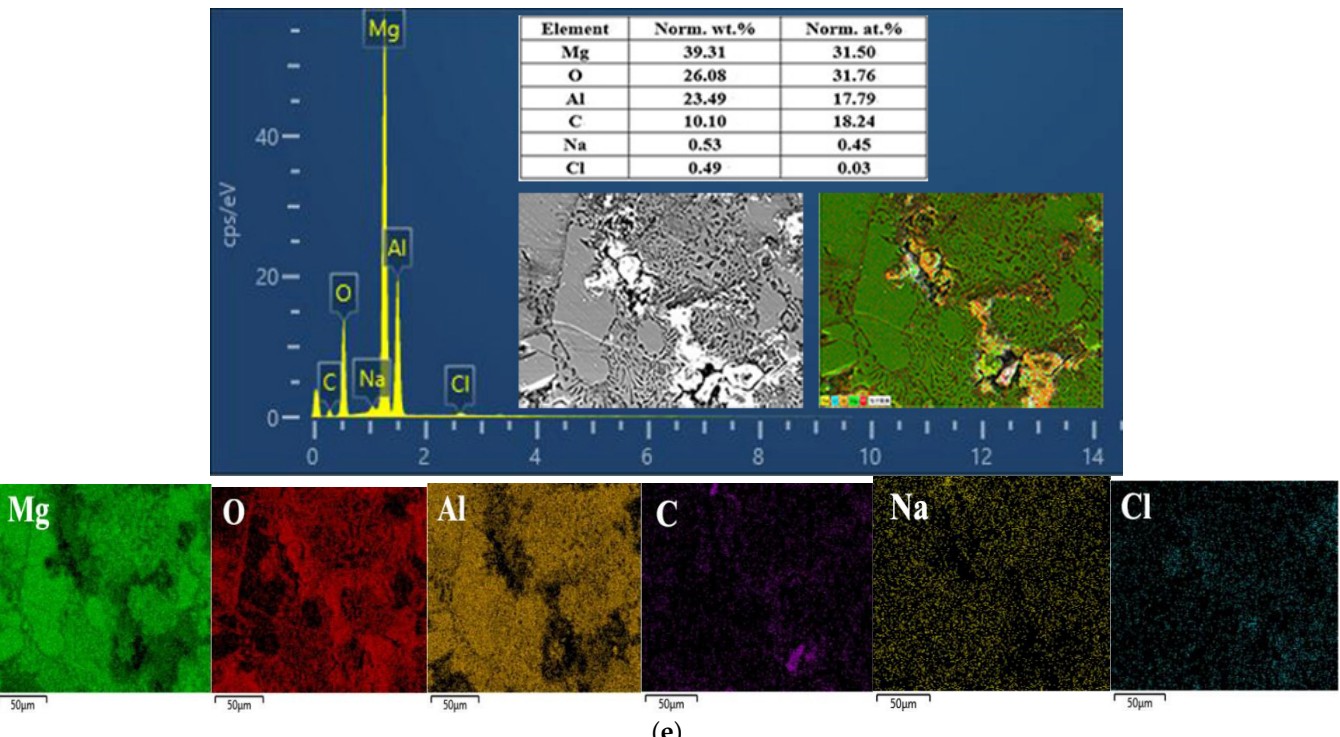

(e)

**Figure 9.** EDS spectrum and the composition and distribution of elements in the (**a**) Al1060, (**b**) Mg1, (**c**) $Mg_2Al_3$ layer, (**d**) $Mg_{17}Al_{12}$ layer, and (**e**) $Mg_{17}Al_{12}$ and Mg-based solid solution layer.

EDS determination results provided a more conclusive figure (Figure 9) on which the overall surface composition of the samples was measured. The elemental composition of the sample surface after electrochemical corrosion can be analyzed more intuitively through surface scanning energy spectrum detection [35]. The EDS spectrum and surface composition indicated the presence of aluminum and oxygen only, along with small amounts of carbon and magnesium on the tested Al1060 surface (see Figure 9a). Most areas of its surface have retained their original integrity. The oxygen was mainly distributed in the vicinity of the corrosion pits. Moreover, the presence of magnesium could be attributed to the fact that it is part of the main composition of Al1060, and the presence of carbon could be attributed to the adsorption of $CO_2$ from the atmosphere. For each sample in Figure 8, Table 5 shows the elemental composition of a surface spot very close to the corrosion products. For example, the EDS analysis of point A shows an Al content of 45.04 wt.% along with an oxygen content of 50.04 wt.%. The corrosion product is, thus, most likely $Al_2O_3$. In the study by Chakradhar et al. [36], the EDAX studies also confirmed that the hazy white areas on the Al block after electrochemical corrosion relate to aluminum oxide, and its formation can be described by Equations (1)–(3):

$$Al^{3+} + 3Cl^- = AlCl_3 \tag{1}$$

$$AlCl_3 + 6H_2O = 2Al(OH)_3 + 3HCl \tag{2}$$

$$2Al(OH)_3 = Al_2O_3 + 3H_2O \tag{3}$$

The EDS spectrum and the surface composition of Mg1 indicated a very complex composition of the surface (see Figure 9b), in which the magnesium content was relatively small, and the oxygen content was quite large. This indicates that the surface was almost completely covered with a thick layer of compounds. Moreover, the EDS analysis showed that box E was mainly composed of 30.25 wt.% magnesium and 56.42 wt.% oxygen (see Table 5), suggesting that the corrosion product is mainly $Mg(OH)_2$. Since the chemical activity of magnesium is lower than that of sodium, it is impossible for the $Mg^{2+}$ ions to be

replaced by $Na^+$ ions in the solution. Thus, it was mainly magnesium that reacted with the aqueous solution, which has taken place as described by Equation (4).

$$Mg + 2H_2O = Mg(OH)_2 + H_2 \tag{4}$$

Figure 9c,d shows the EDS determination results of the $Mg_2Al_3$ layer and $Mg_{17}Al_{12}$ layers, respectively; it can be observed that aluminum and magnesium play a dominant role in their surface composition, oxygen is not widely and evenly distributed on their surface, and oxygen was dominating in the vicinity of the corrosion defects (including pits and cracks) as well as in block- and needle-like corrosion products. However, in the surface composition of the tested $Mg_{17}Al_{12}$ and Mg-based solid solution layer, magnesium, aluminum, and oxygen predominate (see Figure 9f), and oxygen content reaches 26 wt.%, reflecting the existence of many corrosion products on its surface. Moreover, the elemental composition of the corrosion products of these three diffusion layers was mainly magnesium, aluminum, and oxygen (see Table 5). In conclusion, each diffusion layer's corrosion products were not only single component products. There was probably an occurrence of various oxides and hydroxides of Mg and Al, irregularly attached to the surface.

**Table 5.** EDS results of the distinct regions are presented in Figure 8.

| Position | Mole Fraction/% | | | | |
|:---:|:---:|:---:|:---:|:---:|:---:|
| | **Mg** | **Al** | **O** | **Cl** | **C** |
| A | 0.31 | 45.04 | 50.04 | 0.93 | 3.68 |
| B | 30.25 | 0.40 | 58.42 | 3.91 | 7.02 |
| C | 21.96 | 23.48 | 45.62 | 2.68 | 6.26 |
| D | 20.47 | 16.29 | 48.95 | 5.62 | 8.67 |
| E | 22.19 | 12.81 | 58.57 | 1.19 | 4.31 |

After the potentiodynamic electrochemical measurements, by analyzing the surface morphologies, it was possible to conclude that the corrosion damage of the $Mg_{17}Al_{12}$ and Mg-based solid solution layers and of the Mg1 substrate was much more severe than those of the Al1060, $Mg_2Al_3$, and $Mg_{17}Al_{12}$ layers. As a result, the Mg1 substrate was corroded entirely and destroyed, and a thick layer of corrosion products adhered to its surface. Therefore, these corrosion morphology results have verified the results from the corrosion resistance analysis of the polarization curves presented above.

## 4. Conclusions

The present study uses a vacuum diffusion welding process to weld Mg1 and Al1060. In addition, several corrosion resistance experiments were conducted on the resulting Al/Mg intermetallic plates, and a series of microstructural observations could also be made. According to the experimental results, the following conclusions could be drawn:

(1) Vacuum diffusion welding could realize the joining of Mg1/Al1060. The microstructure of the joint was excellent, and uniform diffusion layers were formed at the interface after sufficient diffusion of elements in the material structures. The diffusion layers from the Al side to the Mg side were: $Mg_2Al_3$, $Mg_{17}Al_{12}$, and $Mg_{17}Al_{12,}$ and a Mg-based solid solution layer.

(2) The results of the corrosion immersion tests have demonstrated that the Mg1 substrate was the first to be corroded in a 3.5 wt.% NaCl solution. Severe corrosion damage occurred on this surface after a short period in the solution. The corrosion rates of the Al1060 substrate and the diffusion layers were, thus, slower. The Mg1 substrate, in direct contact with the diffusion layers, acted as an anode in a galvanic cell. It indirectly protected the diffusion layers, which were the latest to be corroded. Among the diffusion layers, corrosion mainly occurred in the combined $Mg_{17}Al_{12}$ and Mg-based solid solution layer.

(3) Linear polarization curves and corrosion morphology analyses also showed that the corrosion resistance of Mg1 was the worst in an aggressive NaCl environment, as compared with the Al1060 substrate and the diffusion layers. It was followed by the combined $Mg_{17}Al_{12}$ and Mg-based solid solution layer. As measured by potential electrochemistry, severe corrosion occurred on the surfaces of these compounds. On the contrary, the $Mg_2Al_3$ and $Mg_{17}Al_{12}$ layers showed excellent corrosion resistance comparable to that of Al1060. The order of corrosion rate of tested samples was Mg1 > $Mg_{17}Al_{12}$ and Mg-based solid solution > $Mg_2Al_3$ > $Mg_{17}Al_{12}$ > Al1060.

In this study, the Mg/Al vacuum diffusion layers were extracted separately for the first try, and the corrosion behavior of each diffusion layer and substrate was studied in depth. However, this research is subject to several limitations. The first is the experimental instruments, the electromagnetic interference generated by alternating current in the electrochemical workstation will have a particular impact on the measurement results. Another limitation concerns the characterization method, more research on electrochemical tests (e.g., electrochemical impedance spectroscopy, cyclic polarization, etc.) can be carried out in the future, and the corrosion resistance of samples can be further discussed based on these tests. However, these limitations will not cause significant prejudice to the current research results and will not affect the research.

**Author Contributions:** Conceptualization, S.Z. and Z.Z.; methodology, Y.D. and D.J.; experiment, S.Z. and Y.D.; data curation, S.Z. and Z.Z; writing—original draft preparation, S.Z.; writing—review and editing, Y.D. and D.J. All authors have read and agreed to the published version of the manuscript.

**Funding:** This research was funded by Scientific Research Funding Project of the Education Department of Liaoning Province (2019LNQN01) and Doctoral Scientific Research Starting Fund of Department of Science and Technology of Liaoning Province (2021-BS-241).

**Institutional Review Board Statement:** Not applicable.

**Informed Consent Statement:** Not applicable.

**Data Availability Statement:** Not applicable.

**Acknowledgments:** This research was supported by the "Liaoning Provincial Laboratory for Special Machining of Complex Workpiece Surfaces" and "Magnesium Alloy Rolling Centre", School of Mechanical Engineering, University of Science and Technology, Liaoning. Moreover, the authors would like to express their gratitude to EditSprings (https://www.editsprings.cn (accessed on 22 August 2022)) for the expert linguistic services provided.

**Conflicts of Interest:** The authors declare no conflict of interest.

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
