# Peer review of "Corrosion Resistance of Mg/Al Vacuum Diffusion Layers"

_coatings, doi:10.3390/coatings12101439_

Round 1

Reviewer 1 Report

The manuscript presents an interesting study about the corrosion resistance of Mg/Al layers obtained by the vacuum diffusion welding process. The layers were characterized by SEM, EDS, OM and linear polarization. However, the paper cannot be processed further in the present form, some comments follow:

Abstract:

The abstract must be improved. Please highlight the novelty of this study. Also, please introduce some quantitative results (one phrase).

Introduction

The introduction section must be improved.

In the last paragraph of the Introduction, please write shortly about the methods used to characterize the layers and highlight the novelty of this study.

Materials and methods

Subsection 2.1.

How was determined the chemical composition of Mg1 and Al1060 steel? or it is given by the manufacturer? Please write this information.

Also, in order to be easier for the reader, please add a table with the acronyms for each sample studied and the difference between them.

From what I can observe, the potentiometric electrochemical measurement used is actually linear polarization. Please be more specific.

Introduce information about the equipment and software used to determine the corrosion resistance. Also, introduce information about the scan rate, potential range etc. Please see an example: DOI: 10.3390/ma13153410

Results and discussion

Figure 4. Introduce figure labels to highlight the zone of interest for the reader.

Table 4. Please calculate and introduce the values of polarization resistance and corrosion rate and discuss them, please see an example: DOI: 10.3390/ma13153410

Add more discussion in this section comparing the results obtained with other studies.

Table 5. Introduce the EDS spectrum and distribution of elements.

The discussion must be improved. If the authors want to characterize the corrosion behaviour using EDS and SEM, please see an example: DOI: 10.3390/ma14010188

Conclusion

The authors must introduce also quantitative results in the conclusion section.

From my point of view, the characterization of samples is poor and the paper is not appropriate to be published in this high-ranked journal. In order to be taken into consideration for publication, I recommend to the authors introduce other tests, like electrochemical impedance spectroscopy, cyclic polarization etc.

Reviewer 2 Report

1. The research significance should be further clarified. If the mechanical properties of joints, as structural materials, are the main index, and intermetallic compounds will have a negative effect on mechanical properties, what is the significance of studying corrosion properties under such conditions.

 2. There are many grammar or writing errors in the manuscript.

 3. How many times was the corrosion immersion tests and potential electrochemical tests repeated?

 4. How to determine the depth of corrosion pits in Fig. 4?

 5. Mechanism analysis needs to be strengthened.

Reviewer 3 Report

This article is scientifically poorly written and lacks innovation for this journal. Therefore, it cannot be accepted in this journal.

Round 2

Reviewer 1 Report

The authors addressed all of my comments. However, the paper needs minor revisor before publishing:

1.     The English writing in the paper also needs to be improved as there are many grammatical errors and incomplete sentences throughout the paper that made it difficult to read.

2.     The conclusion must be improved. Add limitations and suggestions.

Reviewer 2 Report

The authors correctly modified the manuscript for publication. The present manuscript provides a comprehensive and coherent understanding of the work. A minor comment:

EDX and EDS should be used uniformly

Reviewer 3 Report

This paper reports on the corrosion Resistance of Mg/Al Vacuum Diffusion Layers. The article is incomplete in parts, with some explanations and clarifications needed for better understanding. The article is incomplete in parts, with some explanations and clarifications needed for better understanding. I do recommend that this article be published in Journal of Coatings after major revision.

1)      The authors should improve the language of the manuscript carefully to minimize grammatical, and bibliographic errors.

2)      In the abstract section, the main findings of the research should be written numerically and quantitatively.

3)      In the abstract section, refer to the main findings and general conclusions.

4)      Figure 5 has low resolution and should be replaced with higher resolution figure.

5)      Write keywords based on PubMed mesh.

6)      In the introduction and discussion sections, refer to the following articles: I) Evaluation of heavy metal contamination and scaling and corrosion potential in drinking water resources in Nurabad city of Lorestan, Iran. International Journal of Pharmacy and Technology. 2016;8(2):13137-54. II) Investigating of the corrosion and deposition potentials of drinking water sources using corrosion index: a case study of Dehloran. J Chem Pharm Sci. 2016;974:2115.

7)      Similar studies should be mentioned in the introduction of the article.

8)      At the end of the introduction, the novelty of the research should be clearly written.

9)      The discussion section is poorly written and should be improved with more recent studies within the last 5 years.

10)  The strengths and weaknesses of the study should be written.
